# Efficient Point Cloud Object Classifications with GhostMLP

Hawking Lai and K. L. Eddie Law *

Faculty of Applied Sciences, Macao Polytechnic University, Macao 999078, China
* Correspondence: eddielaw@mpu.edu.mo

**Abstract:** Efficient models capable of handling large numbers of data points in point cloud research are in high demand in computer vision. Despite recent advancements in 3D classification and segmentation tasks in point cloud processing, the deep learning PointNeXt and PointMLP models are plagued with heavy computation requirements with limited efficiencies. In this paper, a novel GhostMLP model for point clouds is thus introduced. It takes the advantages of the GhostNet design modules and uses them to replace the MLP layers in the existing PointMLP model. The resulting GhostMLP architecture achieves superior classification performance with lower computation requirements. Compared to the PointMLP, GhostMLP maintains sustainable performance with fewer parameters and lower FLOPs computations. Indeed, it outperforms PointMLP on the ScanObjectNN dataset, achieving 88.7% overall accuracy and 87.6% mean accuracy with only 6 million parameters and 7.2 GFLOPs—about half the resources required by PointMLP. At the same time, GhostMLP-S is introduced as a lightweight version which also outperforms PointMLP in performance. GhostMLP completes faster training and inference with GPU and is the best-performing method that does not require any extra training data in the ScanObjectNN benchmark. Efficient point cloud analysis is essential in computer vision, and we believe that GhostMLP has the potential to become a powerful tool for large-scale point cloud analysis.

**Keywords:** point cloud classification; deep learning; lightweight model

## 1. Introduction

A point cloud is a set of individual data points in a three-dimensional (3D) space. Proper collection of these data points may create an identifiable 3D structure, map, or model. The effective analysis of point cloud data [1], such as semantic segmentation and object detection, is crucial for applications in 3D computer vision, including robotic sensing and autonomous driving. In general, the data points in point cloud contain geometric information, such as geometric coordinates, and other optional information (e.g., colors, normal vectors, etc.). However, the disordered and irregular nature of point cloud data makes analysis challenging. On one hand, the traditional machine learning point cloud clustering methods are feasible for point cloud indoor object detection. One recent example used exponential function [2] to construct a density clustering model according to local density within cutoff distance. On the other hand, the recent advancement in deep learning has been phenomenal. Among them, the PointNet [3,4] was introduced as a solution to overcome the three non-trivial issues of permutation invariance, interactions between points, and transformation invariance. While newer designs, such as PointConv [5], KPConv [6], DeepGCNs [7], and PointTransformer [8], have shown better performance, they are expensive to run, thus limiting their practical uses. Hence, there is a need for lightweight and faster applications while still offering high precision and accuracy.

In 3D computer vision, methods of learning geometric features evolve from landmark MLP models (PointNet, PointNet++) to using convolutions, graphs, and transformer models. Though these newer designs obtain better performance, they mostly execute on expensive commodities, e.g., the high-end GPU card with a large amount of memory. It is atypical to deploy a heavy, slow, but high-performance application in 3D shape

classification, object detection, etc. Instead, a lighter, faster, and relatively high-performance application is what users desire in general. In addition, limited improvement for some newer models can be observed through the experiments of benchmark datasets, such as ModelNet40 [9], ScanObjectNN [10], S3DIS [11], etc. Recently, many latest models such as PointNeXt [12], and PointMLP [13], begin to rethink network design and training strategies. They show that advanced extractors (MLPs, convolutions, graphs, attentions) may not be the main factors to improve overall performance. For instance, PointNeXt performed lots of experiments to assess different training strategies, and their results showed that training strategies are factors in raising the ranks in benchmark tests.

In this paper, we will make two main contributions. Firstly, we shall propose the "GhostMLP" model, a lightweight high-performance MLP network based on the PointMLP. Secondly, we shall explore different training strategies on the ScanObjectNN and ModelNet40 datasets. It is important to notice that factors beyond advanced extractors, such as training strategies, can significantly affect overall system performance. For instance, pre-training and multimodal techniques can improve point cloud analysis, but they may require additional training data. To evaluate the effectiveness of our approach, we provide a detailed comparison with other relevant prior works.

Our primary goal is to address the heavy computation issues in point cloud analysis. This is to trim down the number of parameters in our proposed design, and for the classification performance issue, as depicted in Figure 1, the overall performance among our proposed designs and other existing designs is shown. The GhostMLP, our design, could reach 88.7% overall accuracy (OA) and 87.6% mean accuracy (mAcc) on the ScanObjectNN dataset. That is, the performance of GhostMLP is measured up to that of the PointNeXt framework.

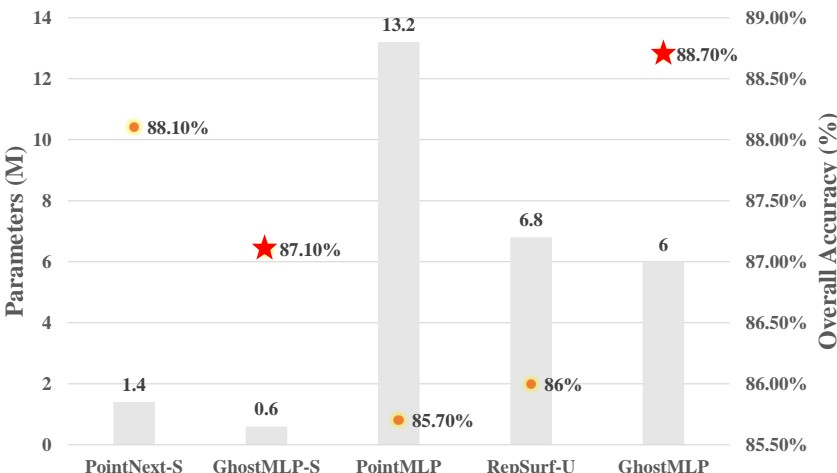

**Figure 1.** Quick view of point cloud object classification performances of different methods in the ScanObjectNN benchmark. Regarding our models, GhostMLP-S and GhostMLP, the former could use minimal parameters to obtain good performance, and the other could achieve the best performance in those models which do not require extra training data.

## 2. Related Works

### 2.1. Deep Learning on Point Cloud

There are two main types of 3D data representations: point-cloud-based and non-point-cloud-based. For the point cloud data, PointNet is the baseline model for processing the point cloud data. Another possible choice is to convert the point cloud to voxel data [14] or multi-view images [15] or other methods; for example, implicit representations [16] could be used to enrich point clouds for better 3D analysis. Traditionally, voxel-based structures are used to study 3D representations, but they require extensive computations to carry out 3D convolutions or run neural networks on 2D multi-view images. On the other hand, multi-view images are a desirable option because they allow for the use of

well-designed 2D neural networks to interpret 3D visual data. It is generally observed that state-of-the-art works can be broadly categorized into two types: those that use pre-training with point cloud data and those that do not. This trend is also reflected in the benchmarks for classification models.

Therefore, classification models can be broadly categorized into two types: those that use pre-training methods with point cloud data and those that do not. Generally speaking, pre-training methods have an advantage in terms of performance, if we do not consider the voting strategy. The use of large models trained with multimodality, which connect computer vision and natural language processing, has allowed models to reach new heights. However, it is also possible to achieve very good performance by training models from scratch without using pre-training. In addition, models trained from scratch tend to have significantly fewer parameters than those that use pre-training methods. Here are some nice SOTA works using pre-training: RECON [17] unifies the data scaling capacity of contrastive models and the representation generalization of generative models to achieve a new state-of-the-art in 3D representation learning on ScanObjectNN. I2P-MAE [18] is an alternative method to obtain high-quality 3D representations from 2D pre-trained models via image-to-point masked autoencoders, which leverage well-learned 2D knowledge to guide 3D masked autoencoding. ULIP [19] is a framework that pre-trains 3D models on object triplets from images, texts, and 3D point clouds by leveraging a pre-trained vision-language model, achieving state-of-the-art performance on 3D classification tasks. These pre-training methods almost aim to improve the representation learning of point clouds for better performance on downstream tasks.

The basic point cloud data structure contains points, and each point is like a pixel in a picture. PointNet [3] is one deep learning neural network model to process the point cloud data. Vanilla PointNet uses an MLP and max pooling to learn a feature. Apart from MLP, there are other approaches using attention (also known as transformers), convolutions, and graphs to produce good benchmark results. Unlike MLPs, these design models are usually complicated, and their approaches are tedious to work on. Hence, most accepted approaches use MLPs, such as the PointNeXt [12], PointMLP, and RepSurf [20], etc. PointNeXt reuses the architecture of PointNet++ [4] with some modifications and uses training strategies to achieve good performance. Then the PointMLP is totally a different architectural MLP design that is not based on the PointNet model, and RepSurf focuses mostly on the point cloud representation. Potentially, a good point cloud representation (triangular RepSurf and umbrella RepSurf) may lead a selected model (e.g., PointNet++) to achieve excellent performance which is comparable to that of the state-of-the-art design. Recent advancements in hyperbolic neural networks (HNN) [21] have led to several studies attempting to project data into the hyperbolic space to improve performance, such as HyCoRe [22], which proposes a new method of embedding point cloud classifier features into hyperbolic space with explicit regularization to capture the inherent compositional nature of 3D objects, achieving a significant improvement in the performance of supervised models for point cloud classification.

In summary, there are various ways to process point cloud data using deep learning models, and the choice of model depends on the specific application and performance requirements. While PointNet is a popular choice for point cloud processing, other models such as PointNeXt, PointMLP, and RepSurf can also achieve excellent results with some modifications.

### 2.2. PointMLP

PointMLP [13] is an extension of the traditional MLP architecture that incorporates residual blocks for better performance on point cloud analysis tasks. The key idea behind PointMLP is that local geometric information may not be as important as previously thought and that simple yet effective methods such as the geometric affine sampling technique can achieve high accuracy without the need for complex geometric feature extraction. This approach not only improves robustness but also allows for faster computation and reduced

model complexity. The use of residual blocks in PointMLP further enhances the network's ability to learn complex representations by enabling it to capture both short-term and long-term dependencies. Additionally, the FPS sampling with the geometric affine method allows for better handling of variations in scale, rotation, and translation, making PointMLP suitable for a wide range of point cloud analysis tasks.

Overall, The main contribution of PointMLP is its use of residual blocks in an MLP framework for point cloud analysis, challenging the notion that local geometric information is necessary for accurate analysis. Additionally, the use of the geometric affine sampling method enhances the robustness and accuracy of the model. PointMLP represents a promising advancement in the field of deep learning for point cloud analysis.

### 2.3. GhostNet

GhostNet [23] is a relatively lightweight deep neural network for image analysis. It believes that the features cloud can be obtained through a lower-cost operation other than a convolution. As shown in GhostNet design, due to feature redundancy, many of them may be more than sufficient to do classification. Its design goal is to make a system lightweight. Hence, many of its follow-up research focuses on more downstream tasks on images [24,25]. The essential component of GhostNet is called the "ghost" modules, which are essentially lightweight versions of standard convolutional neural network (CNN) modules. The ghost modules are designed to use fewer parameters and computations than the regular standard counterparts while still maintaining high accuracy. The idea is to reduce the computational cost while preserving its ability to learn and generalize from data.

### 3. Methodology

In this section, the GhostMLP architecture will be introduced, which leverages the "Sample_and_Group" and "Ghost Set Abstraction" as the main feature extractors in every stage and shown in Figure 2. Architecturally, the GhostMLP is a selective combination of two other deep learning models: the PointMLP and PointNeXt. Through proper training strategies, the goal of GhostMLP is to develop an efficient point cloud object classification neural network, and it can extract more features using fewer resources, further minimizing computation time and running faster during practical use.

Another inspiration for developing an efficient neural network for point cloud object classification comes from the I3D [26]. One contribution of the I3D is to expand a 2D neural network into a 3D network. Conversely, a well-designed 2D neural network can be shrunken into an MLP network to build a point cloud object classification network. However, contraction only may not be sufficient to create an efficient 2D neural network. Therefore, the original GhostNet, designed based on the concept of grouped convolution, is upgraded through the contraction idea to create a network that can be used for point cloud object classification.

Furthermore, it has been noticed that the baseline model of PointMLP does not apply data augmentation to train the neural network. This results in limiting its overall performance improvement. Then through the addition of the PointNeXt component, the extra data augmentation improves the resulting performance of the proposed neural network design. Therefore, a point cloud neural network trained with data augmentation can enhance both the performance of PointMLP and our proposed GhostMLP design as well.

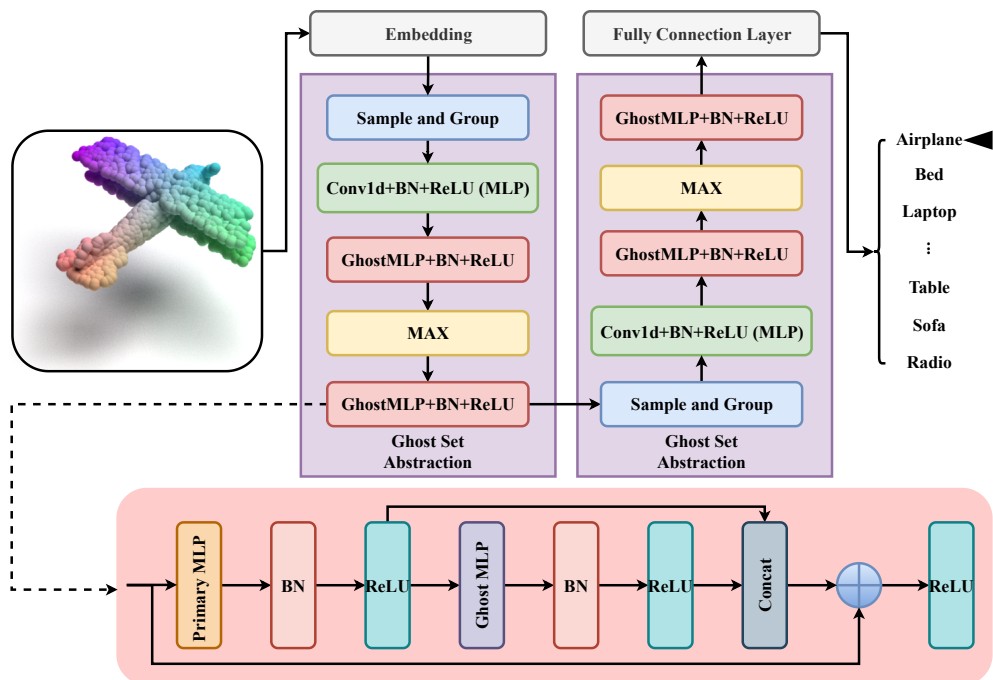

**Figure 2.** Overview of a stage in GhostMLP. For full stages, the neural network mainly comprises an input embedding module and four pairs of ghost set abstraction modules. For the sample and group, a well-designed geometric affine module in PointMLP is applied.

### 3.1. Framework of GhostMLP

**Motivation of a lightweight PointMLP:** Many advanced point cloud extractors are complex. The extractor in PointMLP is effective by adopting a simple framework design. Despite using a simple framework, similar to many other neural network designs, it is slow to run on devices with a small amount of memory and computing resources for the point cloud classification operations. We would like to save resources for subsequent downstream tasks, and we have discovered that the component of GhostNet in our proposed GhostMLP can significantly speed up the computation operations in the network.

**Explore the feature maps in the point cloud.** Unlike many feature maps generated by many neural networks for image processing, PointNet++ creates a different set of feature maps based on the dimensions of sampled points, as shown in Figure 3. These feature maps are not trivial, and redundancies can still be found through cosine similarity calculations. As a result of these findings, the concepts of GhostNet can be applied to point cloud analysis.

**Craft the Ghost block:** In this subsection, out goal is to speed up the networks for point cloud classification by avoid carrying out the heavy downstream computation tasks. To achieve this, we have noticed that significant improvement may be possible through adopting the GhostNet design within GhostMLP. In fact, using GhostNet structures directly also does not meet the goal due to its slow performance in resource (memory and computation) constraint computing systems. Therefore, we propose to create two MLP blocks, denoted *primary_MLP* and *ghost_MLP* (we call them the "primary part" and "ghost part" in the following, respectively), by modifying the GhostNet blocks without duplicating the neural network parameters. We then introduce a residual structure to combine these modified blocks and enable efficient point cloud classification.

Algorithm 1 shows the pseudo code of GhostMLP. It takes three types of inputs: the data, the number of channels, and the ratio. We would like to initially set the *intrinsic_channel* size, which is calculated as a product of the number of input channels and the ratio. Then, we have to set up two MLP components: the *primary_MLP* and *ghost_MLP*. The *primary_MLP* is the main feature extractor, and the *ghost_MLP* is the secondary "cheap operator." Each of these two MLPs consists of a 1D convolutional layer, a batch normalization layer, and a

ReLU activation function. After the initial setting, the input data runs through the *ghost_MLP* module to obtain the first set of features. Afterwards, the *ghost_MLP* is applied to the output of the *primary_MLP* to obtain the second set of features. Finally, the design is to concatenate the two sets of features along the channel dimension and produce the result.

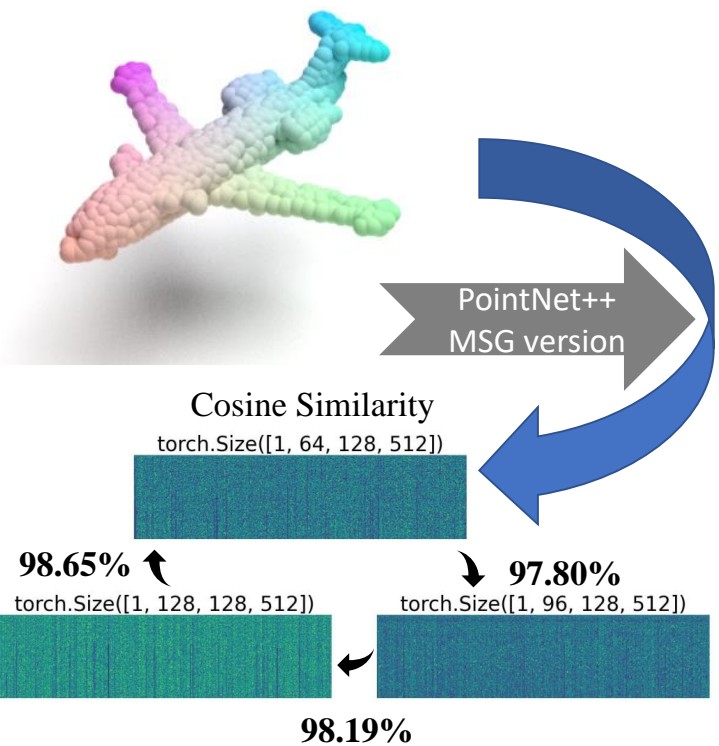

**Figure 3.** Feature maps in PointNet++.

---

**Algorithm 1** GhostMLP in a PyTorch-like style .

---

1: # Primary MLP: main feature extractor
2: # Ghost MLP: cheap operation
3: **function** GHOSTMLP(*data*, *channel*, *ratio*)
4:    **Input:** *data*
5:    **Parameter:** *channel*, *ratio*
6:    **Output:** the feature of data
7:    # Init the channel of primary_MLP:
8:    $intrinsic\_channel \leftarrow channel \times ratio$
9:    # Params: channel, intrinsic_channel
10:    $primary\_MLP \leftarrow Conv1d + BatchNorm1d + ReLU$
11:    # Init the channel of ghost_MLP:
12:    $ghost\_channel \leftarrow intrinsic\_channel$
13:    # Params: intrinsic_channel, ghost_channel
14:    $ghost\_MLP \leftarrow Conv1d + BatchNorm1d + ReLU$
15:    # Forward
16:    $feature1 \leftarrow primary\_MLP(data)$
17:    $feature2 \leftarrow ghost\_MLP(data\_feature1)$
18:    $out \leftarrow torch.cat([feature1, feature2], dim = 1)$
19:    **return** *out* # The feature of data
20: **end function**

---

In summary, the GhostMLP algorithm is to apply the input data through the two MLP components (*primary_MLP* and *ghost_MLP*) in order to efficiently extracts feature maps. Then by concatenating the generated features from these MLPs, the design obtain more response results with fewer parameters than the traditional MLP designs.

**Structure of GhostMLP:** Our proposed GhostMLP architecture takes both the positive designs in PointNet++ and PointMLP models. Units such as the geometric affine for sampling and grouping are adopted. In Figure 2, the forward pass of GhostMLP with only one stage is shown. The geometric affine design in PointMLP is adopted in our GhostMLP for the robustness in performance. In fact, we can adapt to different geometric structures in different local regions in order to sample and group the point cloud data.

The following two equations (Equations (1) and (2)) are used in defining the structures of the *Geometric_Affine* module. The first equation computes the output of the *Geometric_Affine* module, and the second equation computes the standard deviation of the feature maps of a local region.

$$Geometric\_Affine = \alpha \odot \frac{\{f_{i,j} - f_i\}}{\sigma + \epsilon} + \beta \tag{1}$$

$$\sigma = \sqrt{\frac{1}{k \times n \times d} \sum_{i=1}^{n} \sum_{j=1}^{k} (f_{i,j} - f_i)^2} \tag{2}$$

Equation (3) represents the GhostMLP module in a neural network stage, which consists of two ghost set abstraction blocks: *Geometric_Affine*, and a series of MLPs. Firstly, the input point cloud data is grouped based on geometric affinity using the *Geometric_Affine* module with learnable parameters. After receiving data from the *Geometric_Affine* module, the neural network needs an operation to reshape the data so that it can be passed through the subsequent MLP layers. Third, the data is sent to the module shown in Algorithm 1 to perform a MAX operation to process the features in each local region.

So far, GhostMLP has a design similar to PointNet, but PointNet's main contribution lies in its ability to learn features of point clouds, given that multi-layer perceptrons (MLPs) struggle with the irregularity of point clouds. However, if we input features that have already been well-learned by MLPs into another MLP, then the challenge brought by point clouds for MLPs no longer exists. Hence, there is one module shown in Algorithm 1 to learn the features on the bottom of the ghost set abstraction. In summary, our model could be shown as follows:

$$Ghost\_Set\_Abstraction = \Phi_{Ghost}^2(\max(\Phi_{Ghost}^1(\Phi(Geometric\_Affine(x))))) \tag{3}$$

where $\Phi_{Ghost}$ represents the MLP block in Algorithm 1, and $\Phi$ represents normal MLP, and $x$ represents the input data. We crafted two types of neural networks named: GhostMLP and GhostMLP-S, the deeper neural network (40 layers) is named as GhostMLP, and the model with 28 layers is for GhostMLP-S.

### 3.2. Rethinking Training Strategies

We acknowledge that some training strategies may limit the overall system performance of our GhostMLP framework. The training operations of the PointMLP model do not include any data augmentation techniques, which is similar to other existing models. We can look at the data augmentation and training strategies used in PointNext as a reference to potentially improve the accuracy and robustness of our GhostMLP framework. If these training strategies and techniques have positive impacts on GhostMLP, such as on better accuracy or robustness, we should thoroughly consider if these strategies can be applied in the design of our GhostMLP framework.

### 3.2.1. Data Augmentation

Upon reviewing the source code of PointMLP, we noticed that it mainly uses translation as the only data augmentation technique. However, incorporating different data augmentation techniques can often improve the performance of neural networks. Therefore, we aim to explore and incorporate various data augmentation techniques in our GhostMLP framework.

### 3.2.2. Training Strategies

To optimize a point cloud classification model, there are two algorithms available: SGD with momentum in PointMLP, and AdamW in PointNext. These techniques have been extensively tested and evaluated in various scenarios. After analyzing the results from PointNext and conducting our own experiments in Section 4.2, we intend to implement AdamW instead of SGD in our GhostMLP framework for the ScanObjectNN dataset. However, it is essential to acknowledge that random factors in the dataset (e.g., ModelNet40) may still impact the performance of our GhostMLP framework. Therefore, in such cases, using SGD can help improve our model's performance.

## 4. Experiments and Result Analysis

In this section, the performance benchmarks of the GhostMLP models for 3D shape classifications and segmentation on three datasets: ModeNet40, ScanObjectNN, ShapeNet-Part, and Oakland 3D are evaluated. The model's performance is assessed on two main issues: (1) the extent to which performance can be sustained or improved while removing certain parameters for different components in the model, and (2) the techniques that can be used to increase/decrease the model's performance.

To conduct the experiments, the GhostMLP models were trained using a cross-entropy loss function with label smoothing and either SGD or AdamW for different tasks. The hyperparameter settings for each result were indicated accordingly. The default batch size was set to 32, but in the case of a memory-constrained GPU (e.g., 12 GBytes RTX3080ti), a batch size of 16 was used. To accelerate the rate of sampling a point cloud, a CUDA accelerating toolkit provided in the PointMLP repository was utilized.

### 4.1. How Lightweight a GhostMLP Is

In the experiments conducted, the GhostMLP and GhostMLP-S models were trained with the same set of hyperparameters and compared against the PointMLP model using the ModelNet40 benchmark. The benchmark consisted of 40 CAD models, with 9843 models used for training and 2468 models used for testing. The evaluation metrics used were mean accuracy (mAcc) and overall accuracy (OA), and the number of parameters and FLOPs were also measured to determine if the models were lightweight enough.

The measured evaluation metrics are the usual mean accuracy (mAcc) and overall accuracy (OA). Another important goal of the experiments was to determine "whether it is lightweight enough." Therefore, the number of parameters and FLOPs were also measured for model evaluations. The training details were almost identical to those in training PointMLP, except that the training speed was different due to the use of the RTX3090 card. Although the performance of 3090 was similar to 3080ti, it came with more GPU memory.

Table 1 shows that GhostMLP performs similarly to PointMLP in terms of mAcc and OA, but outperforms PointMLP in terms of completion times and fewer computations needed. GhostMLP has about half the number of parameters and FLOPs compared to PointMLP, leading to shorter training and testing durations. GhostMLP is optimized for devices with limited computing resources and was found to complete tests faster than PointMLP when run on an Intel I9-9900 CPU.

**Table 1.** Performance of GhostMLP upon comparing to other approaches using identical training configuration. The unit of speed is samples per second. (*: the result is obtained through RTX3090; Numbers, in light cyan background color, are retrieved from PointMLP [13].)

| Methods | Inputs | mAcc (%) | OA (%) | Parameters | FLOPs | Train Speed | Test Speed |
|---|---|---|---|---|---|---|---|
| PointNet | 1 k points | 86.0 | 89.2 | | | 223.8 | 308.5 |
| PointNet++ | 1 k points | | 90.7 | 1.41 M | | 223.8 | 308.5 |
| PointMLP w/o vote | 1 k points | 91.3 | 94.1 | 12.6 M | 14.6 G | 92 * | 205.6 * (7.7 w/. CPU) |
| PointMLP-elite w/o vote | 1 k points | 90.9 | 93.6 | 0.7 M | 0.8 G | 328 | 822.6 (31 w/. CPU) |
| GhostMLP w/o vote | 1 k points | 91.5 | 93.9 | **6.0 M** | **7.2 G** | **118.5** | **308.5 (11 w/. CPU)** |
| GhostMLP-S w/o vote | 1 k points | 90.6 | 93.3 | **0.6 M** | **0.7 G** | **378** | **823 (32.5 w/. CPU)** |

The experiments aimed to evaluate the inference speed of the models based on how many samples per second they could process, with a higher value indicating a better model. GhostMLP was found to execute faster than PointMLP in general, and GhostMLP-S was similar to PointMLP-elite.

*4.2. Classification on ScanObjectNN and ModelNet40*

This section presents an evaluation of the classification performance of the GhostMLP model on the ScanObjectNN and ModelNet40 datasets. The ScanObjectNN dataset, which consists of 2880 objects from 15 categories (2304 training sets and 576 test sets) collected from the real world, presents more complex and challenging scenarios than many traditional classification datasets auch as ModelNet40. Thus, the performance of models trained by ScanObjectNN is expected to be more stable. By utilizing data augmentation and the design of Figure 2, the GhostMLP achieved an overall accuracy (OA) of **88.723%** and a mean accuracy (mAcc) of **87.642%** on ScanObjectNN, without using multimodal or pre-trained models. Notably, GhostMLP outperforms PointMLP on ScanObjectNN by 3% in OA and 3.2% in mAcc.

The GhostMLP was configured with specific parameters, including cosine decay = 0.05, learning rate = $2 \times 10^{-3}$, cross-entropy loss with label smoothing (set to 0.3) [27], AdamW, and seed = 3407 [28] to ensure reliable and predictable results. Table 2 displays the performance comparison of the GhostMLP and several state-of-the-art models on both ScanObjectNN and ModelNet40 datasets. The GhostMLP achieved competitive performance with GhostMLP-S, which utilized a smaller model with fewer parameters.

ModelNet40, which contains 9843 training and 2468 testing data across 40 categories, is a popular dataset in point cloud classification. However, its simplicity poses a challenge for classification experiments and may result in a wide range of performance results when the same code is run multiple times. This issue has been widely discussed in the GitHub Issue of PAConv [29]. Therefore, results for some state-of-the-art designs can vary.

**Table 2.** Performance of classification on ScanObjectNN and ModelNet40. For ModelNet40, no voting was done for all the results here, unless it is noted.

| Methods | ScanObjectNN | | ModelNet40 | | Parameters |
|---|---|---|---|---|---|
| | mAcc(%) | OA(%) | mAcc(%) | OA(%) | |
| PointNet [3] | 63.4 | 68.2 | 86.0 | 89.2 | - |
| PointNet++ [4] | 75.4 | 77.9 | - | 90.7 | 1.41 M |
| PointMLP [13] | 84.4 | 85.7 | 91.3 (+0.1 w/ vote) | 94.1 (+0.4 w/ vote) | 12.6 M |
| PointMLPElite | 82.6 | 84.4 | 90.7 (+0.2 w/ vote) | 93.6 (+0.4 w/ vote) | 0.7 M |
| PointNeXt [12] | 86.8 | 88.2 | 91.6 | 94.0 | 1.6 M |
| HyCoRe [22] | 87.0 | 88.3 | 91.9 | 94.5 | - |
| GhostMLP-S | **85.8 ± 0.2** | **87.1 ± 0.3** | 90.9 (+0.1 w/ vote) | 93.3 (+0.6 w/ vote) | **0.6 M** |
| GhostMLP | **87.2 ± 0.4** | **88.4 ± 0.3** | 91.6 (+0.4 w/ vote) | 93.7 (+0.3 w/ vote) | 6.0 M |

In summary, the GhostMLP demonstrated superior classification performance on ScanObjectNN and competitive results on ModelNet40 datasets compared to other state-of-the-art models. These results highlight the effectiveness of the proposed GhostMLP model design and training strategy.

### 4.3. Part Segmentation on ShapeNet and Classification on MLS Data

To demonstrate the effectiveness of GhostMLP in other domains, we will use part segmentation, which is the task of classifying each point in a point cloud with its corresponding part or category. By performing part segmentation on point clouds, we can obtain detailed segmentations and labels for different parts of the point cloud, which can help us better understand the point cloud data and support higher-level tasks such as object recognition and pose estimation.

### 4.3.1. ShapeNet

In this section, we evaluate the part segmentation performance of GhostMLP on the ShapeNetPart [30] dataset, a common and popular dataset for part segmentation tasks. The GhostMLP model was trained using the LeakyReLU activation function, AdamW optimizer without weight decay, CosineAnnealingLR scheduler, and a total of 600 epochs. Data augmentation was not used due to the nature of part segmentation.

For most object categories in part segmentation, the GhostMLP method achieved similar or better performance compared to two other methods (PointNet++ and PointMLP) on the ShapeNetPart dataset, as shown in Table 3. Specifically, GhostMLP achieved the best accuracy in categories such as backpack, earphone, guitar, chair, table, and knife. In terms of instance segmentation, GhostMLP achieved an accuracy of 86.1%, the same as PointMLP but better than PointNet++. The results demonstrate the effectiveness of the GhostMLP model in 3D object recognition tasks and suggest that the model can learn well with fewer features. Meanwhile, GhostMLP also exhibits efficient performance during training, with the ability to infer 154 models per second.

**Table 3.** Performance of ShapeNet. Instance means instance average IoU, and Class means class average IoU. For every object, the evaluation metric is IoU.

| Methods | ShapeNetPart (Unit: IoU) | | | | | | | |
| | Instance | aero | bag | cap | car | chair | earphone | guitar | knife |
|---|---|---|---|---|---|---|---|---|---|
| PointNet++ | 85.1 | 82.4 | 79.0 | 87.7 | 77.3 | 90.8 | 71.8 | 91.0 | 85.9 |
| PointMLP | 86.1 | 83.4 | 83.3 | 87.4 | 80.5 | 90.3 | 78.1 | 92.1 | 88.0 |
| GhostMLP | 86.1 | 84.5 | 87.0 | 88.3 | 80.6 | 90.3 | 81.3 | 92.0 | 88.6 |

| Methods | ShapeNetPart (Unit: IoU) | | | | | | | |
| | Class | lamp | laptop | moto | mug | pistol | rocket | stake board | table |
|---|---|---|---|---|---|---|---|---|---|
| PointNet++ | 81.9 | 83.7 | 95.3 | 71.6 | 94.1 | 81.3 | 58.7 | 76.4 | 82.6 |
| PointMLP | 84.5 | 82.5 | 96.2 | 77.5 | 95.7 | 85.3 | 65.7 | 83.3 | 84.3 |
| GhostMLP | 85.0 | 82.1 | 96.0 | 77.6 | 95.0 | 84.3 | 64.2 | 83.7 | 84.0 |

Figure 4 presents the results of using GhostMLP for point cloud part segmentation. Through visualization, we found that GhostMLP is not sensitive to end-part objects, which leads to average performance in this task. However, we also noticed that GhostMLP performs well in boundary discrimination, such as the clear segmentation relationship between the wings and fuselage of the airplane in the Figure 4a,e (similar observations also could be found in other subfigures). Therefore, the advantages of GhostMLP are also demonstrated.

In conclusion, GhostMLP demonstrates effective performance in point cloud part segmentation, achieving similar or better performance compared to other methods on

the ShapeNetPart dataset while also exhibiting efficient training performance and good boundary discrimination.

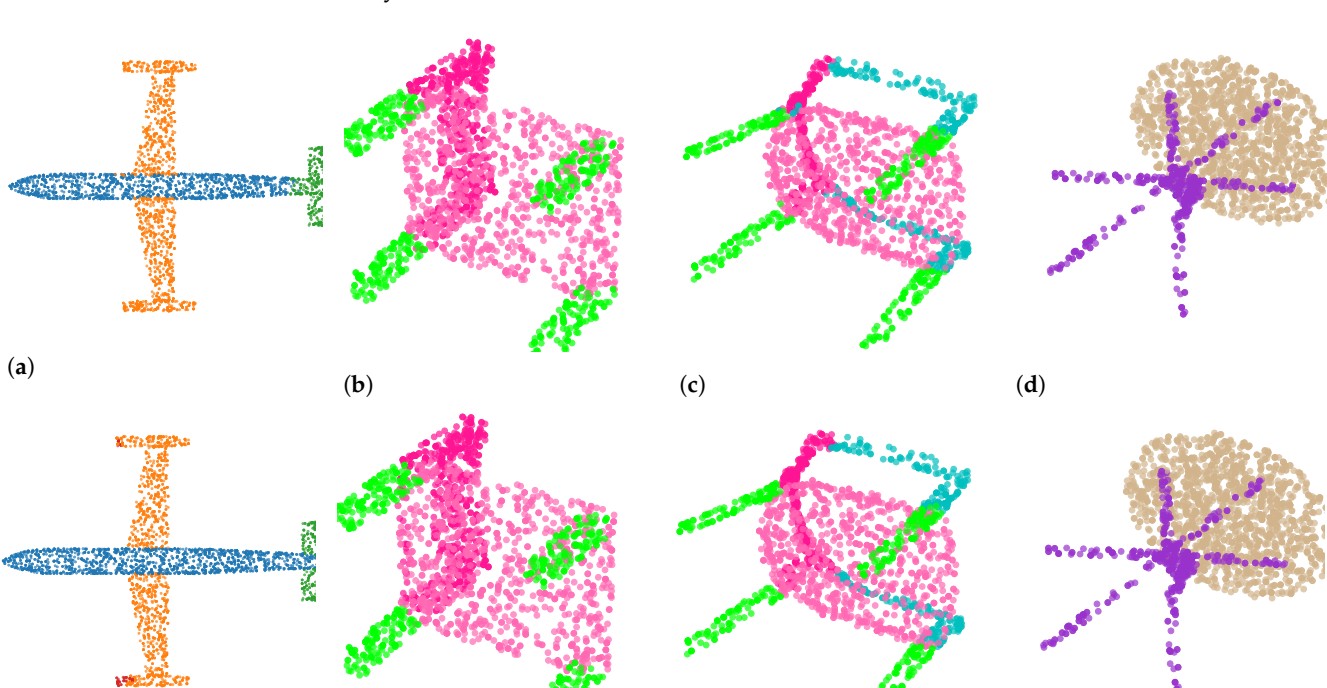

**Figure 4.** Visualization of part segmentation on the ShapeNet-Part dataset. (**a**) Ground truth of airplane; (**b**) ground truth of chair; (**c**) ground truth of chair; (**d**) ground truth of desk; (**e**) prediction of airplane; (**f**) prediction of chair; (**g**) prediction of chair; (**h**) prediction of desk.

### 4.3.2. Oakland MLS Dataset

GhostMLP has promising applications in various remote sensing datasets from a remote sensing perspective, including mobile lidar scans (MLS) or airborne lidar scans (ALS) data. To demonstrate its versatility, we utilized GhostMLP in the Oakland dataset [31,32], which we customized to satisfy the input of GhostMLP. The Oakland dataset is comprised of three-dimensional MLS point clouds captured using a Velodyne HDL-64E LIDAR sensor.

The process of customizing the dataset involved several steps:

1.  Formatting the dataset to be compatible with GhostMLP. We found that the Oakland dataset was formatted by $x, y, z, label, confidence$. Since the confidence values were almost always 2, we assumed that confidence was always true and removed this property. We allocated 5 types of colors to the labels since Oakland only had 5 classes. After formatting, the Oakland dataset was structured as $x, y, z, R, G, B, label$.
2.  Classifying according to the area and dividing large point clouds into smaller ones. Although inputting the entire scene is a popular method for semantic segmentation, we recommend dividing large point clouds into smaller ones because we have labels for each point cloud. Therefore, a point cloud with approximately 100k points will be shuffled and divided into multiple point clouds with 4k points.
3.  Dividing the dataset into training, testing, and validation sets. Finally, we shuffled the 4k point clouds and divided them into training, testing, and validation sets with a ratio of 7:2:1.

As shown in Figure 5, GhostMLP can also perform well on other lidar point clouds. Meanwhile, It can be observed that the buildings are colored purple, the ground is red, the vegetation is yellow, and the poles are green, so they are almost perfectly classified.

Moreover, this visualization, which consists of 280,000 points, only takes GhostMLP 3.2 s to infer, which is fast enough.

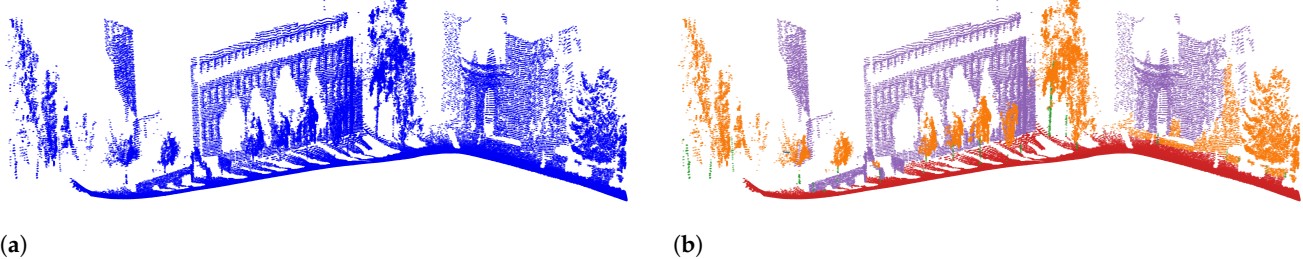

(**a**)                          (**b**)

**Figure 5.** Visualization of classification results on the Oakland dataset. (**a**) Raw data of Oakland dataset. (**b**) Classification results of Oakland dataset.

### 4.4. Interpretability Analysis of GhostMLP

This section presents research on the impact of data augmentations and GhostMLP modules on the GhostMLP framework's performance, along with a loss landscape visualization of GhostMLP and GhostMLP-S.

#### 4.4.1. Ablation Studies: Data Augmentations

In this section, our experiments demonstrate that certain data augmentations can improve the performance of the GhostMLP framework. However, not all data augmentations provide consistent improvements. We conducted a series of experiments to determine the best combination of data augmentations for GhostMLP, and the results are presented in Table 4.

**Table 4.** Performance with different data augmentations.

| Methods in ScanObjectNN | Mean Accuracy (%) | Overall Accuracy (%) | Buff or not |
|---|---|---|---|
| GhostMLP-S (Baseline) | $83.2 \pm 0.2$ | $85.0 \pm 0.3$ | |
| +epoch==600 (slightly called: epoch) | $83.2 \pm 0.2$ | $84.8 \pm 0.4$ | ✗ |
| +rotation +epoch | $85.0 \pm 0.2$ | $86.7 \pm 0.1$ | ✓ |
| +point dropout +epoch | $84.5 \pm 0.2$ | $86.3 \pm 0.2$ | ✓ |
| +scale +epoch | $82.7 \pm 0.4$ | $84.2 \pm 0.3$ | ✗ |
| +rotation + point dropout +epoch | $85.8 \pm 0.2$ | $87.1 \pm 0.3$ | ✓ |
| +rotation + point dropout + scale +epoch | $85.8 \pm 0.2$ | $87.1 \pm 0.3$ | ✓ |

We began by exploring if increasing the number of epochs could improve the performance of the model. However, we found that just increasing the epochs did not result in any noticeable improvement. Next, we evaluated the effects of three common data augmentations used in point cloud representation learning: rotation, random point cloud dropout, and random scaling of the point cloud. We added one type of data augmentation at a time and set the epoch to 600. Our findings indicate that both rotation and random point cloud dropout can improve the performance of the model, but random scaling of the point cloud cannot. The reason for this failure may be due to the use of *PB_T50_RS*, which is one of the most challenging and commonly used variants of ScanObjectNN, and the point cloud data in this variant has already been rescaled.

In the final step, we tested different combinations of these three data augmentations to determine if they could further improve the performance of the model. We observed that adding data augmentations generally had a positive impact on the model performance, especially when using rotation and random point cloud dropout. Even adding random scaling of the point cloud can be helpful when combined with other data augmentations.

From the Table 4, it can be seen that adding rotation and random point cloud dropout individually improved the mean and overall accuracy of the model. Adding random scaling of the point cloud, however, decreased the mean and overall accuracy. When combining the three data augmentations, the performance of the model improved significantly, with both combinations resulting in an increase in mean and overall accuracy.

Overall, our experiments indicate that reasonable data augmentations can improve the performance of the GhostMLP framework, including GhostMLP-S and etc., based on the ScanObjectNN dataset. However, it is important to note that not all data augmentations are effective, and the choice of data augmentation should be carefully considered.

### 4.4.2. Ablation Studies: Modules in GhostMLP

The equation given in (3) outlines the ghost set abstraction process, which involves applying a geometric affine transformation on the input data $x$, followed by two MLPs $\Phi^1_{Ghost}$ and $\Phi^2_{Ghost}$, and then taking the maximum of the output to obtain the final result. This framework can significantly reduce computation, and thus it is necessary to test whether applying both $\Phi^1_{Ghost}$ and $\Phi^2_{Ghost}$ would lead to a loss in performance.

To conduct the test, we trained four types of neural networks on the ScanObjectNN benchmark with the hyperparameters set as in Section 4.2. The results, as shown in Table 5, indicate that the performance of the framework that combines both $\Phi^1_{Ghost}$ and $\Phi^2_{Ghost}$ is comparable to the frameworks that include only one or none of them. Furthermore, using the framework that combines $\Phi^1_{Ghost}$ and $\Phi^2_{Ghost}$ leads to better results and has lower parameter and FLOP counts compared to the other frameworks.

**Table 5.** Ablation study for contributions of $\Phi^1_{Ghost}$ and $\Phi^2_{Ghost}$. (Model: GhostMLP-S).

| $\Phi^1_{Ghost}$ | $\Phi^2_{Ghost}$ | OA (%) | mAcc (%) | Parameters (M) | FLOPs |
|---|---|---|---|---|---|
| ✓ | ✓ | $87.3 \pm 0.3$ | $85.8 \pm 0.3$ | 0.597 | 0.725G |
| ✗ | ✓ | $87.1 \pm 0.2$ | $86.0 \pm 0.3$ | 0.637 | 0.844G |
| ✓ | ✗ | $87.2 \pm 0.2$ | $85.7 \pm 0.4$ | 0.637 | 0.730G |
| ✗ | ✗ | $87.2 \pm 0.2$ | $85.8 \pm 0.4$ | 0.683 | 0.849G |

### 4.4.3. Ablation Studies: Network Depth

In this study, we aim to investigate the impact of network depth on the performance of our proposed framework. To determine the depth of our network, we use the following equations:

$$L = 1 + \sum_{i=1}^{S}(1 + 2 \times r_i \times (1 + 1)) + 3 \tag{4}$$

$$S = len(r) \tag{5}$$

Here, $L$ represents the number of layers in the network. The first layer is the embedding layer, and $S$ represents the current stage of ghost set abstraction operations in the network. Each ghost set abstraction operation includes one MLP for data reshaping and two GhostMLP operations. Each GhostMLP consists of a primary MLP and a ghost MLP, which are repeated r times. Two series of GhostMLP operations are performed in one stage of the ghost set abstraction. Finally, the classifier layer contains three layers. By using this Equation (4), the depth of our network can be calculated and analyzed for its impact on performance.

Table 6 shows the results of the ablation study conducted to investigate the impact of network depth on the performance of the proposed framework. Three different network depths were tested, with the number of layers ranging from 24 to 40. The table presents the repeat parameter used for each depth, the calculated number of layers, and the corresponding mean accuracy (mAcc) and overall accuracy (OA). The results indicate that increasing the network depth improves the performance of the proposed framework, as shown by

the increase in both mAcc and OA with increasing depth. Specifically, the network with a repeat parameter of $[2, 2, 2, 2]$, resulting in 40 layers, achieved the highest mAcc and OA, indicating that a deeper network is more effective for the given task.

**Table 6.** Performance of two types of network depth. (Dataset: ScanObjectNN).

| Repeat | Depth | mAcc (%) | OA (%) |
|---|---|---|---|
| $[1, 1, 1, 1]$ | 24 layers | $81.2 \pm 0.2$ | $83.4 \pm 0.3$ |
| $[1, 1, 2, 1]$ | 28 layers | $85.8 \pm 0.2$ | $87.1 \pm 0.3$ |
| $[2, 2, 2, 2]$ | 40 layers | $87.2 \pm 0.4$ | $88.4 \pm 0.3$ |

The results indicate a significant improvement in performance with an increase in network depth. However, achieving 40 layers, the improvement in performance becomes less significant. Therefore, it can be concluded that increasing the depth of the network can contribute to the performance of the GhostMLP framework, but with diminishing returns beyond a certain point.

### 4.4.4. Loss Landscape

In this section, we present the loss landscapes [33] of the GhostMLP framework, which provides a means of interpreting the model's behavior. However, it is important to note that a smooth and flat landscape does not guarantee optimal performance, as other factors such as overfitting, underfitting, and optimization algorithms can also affect the network's performance. Nonetheless, a good landscape is generally considered desirable for a neural network.

As shown in Figure 6, two loss landscapes of the GhostMLP framework are displayed. Figure 6a,b illustrate a smooth and flat landscape characterized by few local minima and a broad global minimum. These features indicate that the network is less likely to obtain stuck in local minima and more likely to converge to the global minimum, resulting in improved training and test performance.

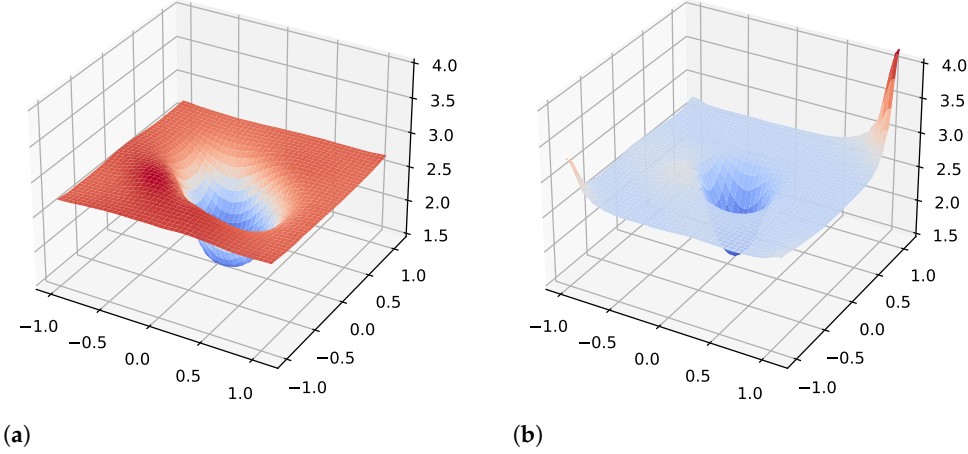

(**a**)           (**b**)

**Figure 6.** The loss landscape of GhostMLP and GhostMLP-S. (Dataset: ScanObjectNN). (**a**) Loss landscape of GhostMLP. (**b**) Loss landscape of GhostMLP-S.

### 5. Discussion

In this section, we will compare GhostMLP with other designs. GhostMLP is a model that trains from scratch and does not rely on large, multi-modal, pre-trained models. Although it cannot outperform them, it achieves satisfactory performance evaluations for most applications that prioritize efficiency over a small difference in accuracy. Table 7 summarizes the architecture comparison of GhostMLP with other models that train from scratch.

**Table 7.** Architecture comparison on baseline and state-of-the-art works.

| Methods | Group and Sample Methods | Feature Spaces | Feautre Extraction |
|---------|--------------------------|----------------|--------------------|
| PointNet [3] | FPS | Euclidean | MLP |
| PointNet++ [4] | FPS | Euclidean | MLP |
| Point Transformer [8] | FPS | Euclidean | Attention |
| PointNeXt [12] | FPS | Euclidean | MLP |
| HyCoRe [22] | Geometric Affine | Hyperbolic | MLP |
| PointConT [34] | FPS | Euclidean | Attention |
| GhostMLP | Geometric Affine | Euclidean | MLP |

We observe that state-of-the-art works such as Point Transformer and PointConT achieve good performance due to their advanced feature extraction methods. Similarly, PointNeXt improves its performance by using better training strategies. HyCoRe and GhostMLP are both extensions of PointMLP. HyCoRe modifies the projection to hyperbolic space without changing the basic feature extraction design, while GhostMLP improves its performance by using efficient and effective feature extraction methods and training strategies.

However, GhostMLP is not flawless. It sacrifices some performance in the benchmark racing because the ghost set abstraction only uses part of the features to learn to be a lightweight neural network. This is a limitation that can be addressed in future work. Another possible direction for improvement is to explore the use of hyperbolic space instead of Euclidean space for feature representation.

## 6. Conclusions

In this paper, we proposed the GhostMLP model for efficient point cloud object classifications. GhostMLP is a novel neural network architecture that introduces a new residual MLP module which significantly reduces the number of parameters without compromising the classification performances. The GhostMLP is evaluated and validated with the two popular point cloud classification datasets: ScanObjectNN and ModelNet40. The results show that GhostMLP and its variant GhostMLP-S achieve competitive performance and fast inference compared with other state-of-the-art methods and their baseline. We also applied GhostMLP to other tasks such as part segmentation on ShapeNet-Part and classification on mobile lidar scan (MLS) data, demonstrating its versatility and efficiency for different point cloud recognition applications. On running the ablation study, we discovered that the effectiveness of GhostMLP is attributed to the combination of ghost modules, data augmentations, and deeper network architecture. Through analyzing the loss landscape of GhostMLP, it is indeed smooth and close to a flat surface. This can explain the outstanding performance in its superior generalization ability. However, GhostMLP still has some limitations that we plan to address in future work. For instance, we need to perform a more comprehensive analysis of the hyperparameters and the sensitivity to input perturbations to improve the interpretability of GhostMLP. We also need to explore ways to enhance the robustness and generalization of GhostMLP, such as designing a more effective ghost set abstraction module and applying hyperbolic neural networks for feature representation.

Overall, our findings suggest that GhostMLP offers a promising direction for efficient point cloud object classification and has the potential to be applied to a wide range of point-cloud-related tasks.

**Author Contributions:** Conceptualization, H.L.; Methodology, H.L.; Software, H.L.; Writing—original draft, H.L.; Writing—review & editing, K.L.E.L.; Supervision, K.L.E.L. All authors have read and agreed to the published version of the manuscript.

**Funding:** The authors would appreciate the research funding support (#RP/ESCA-09/2021) from Macao Polytechnic University for the project.

**Data Availability Statement:** All data are open public data. ModelNet40 could be downloaded here, ScanObjectNN could be downloaded here, ShapeNet could be downloaded here, Oakland 3-D Point Cloud Dataset could be downloaded here.

**Conflicts of Interest:** The authors declare no conflict of interest.

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
