# Peer review of "Efficient Point Cloud Object Classifications with GhostMLP"

_remotesensing, doi:10.3390/rs15092254_

Round 1
Reviewer 1 Report
This is an interesting paper describing Point Cloud Object Classifications.
I would like to see this study published, however in its current form the paper is not ready for publication.
While the results are fairly well-elaborated, the text requires minor additions.
Some main points are detailed below.
· • Section 2. is lacking in literature and reference. The topics covered in the article are very popular and you can find a lot of interesting and valuable articles, e.g. in RS: https://www.mdpi.com/2072-4292/15/1/61
· • The article has a disturbed content structure. Readable parts are missing: Methodology, Results, Conclusions, and Discussion.
· • The conclusions section contains no conclusions. Lines 384 - 387 simply reiterate the results and do not elaborate on their significance, these are already mentioned in the results section.
· • In general, the conclusions should be entirely rewritten. Also, address the ways in which the research may be used in other contexts, and how, more specifically, future work can improve upon the current results.
· • Finally, most of the conclusions are observations they should integrate and highlight strengths and weak points of the proposed method possibly referring also to a comparison with the state of the art which could be done in the Discussion Section (that is currently missing). I think that this correction can substantially increase the interest of the remote sensing community.
Author Response
Dear Reviewer #1,
Thank you for taking the time to review our paper and for providing us with your valuable feedback. We appreciate your interest in our work and your constructive comments, which have helped us to improve the quality of our paper. We have carefully revised our paper based on your suggestions and have made the following changes:
Firstly, we have added more literature review and references in Section 2, particularly on the state-of-the-art works in the benchmark of ScanObjectNN. We have analyzed their strengths in Section 2 and compared them with our proposed method in the Section discussion. We have also included the paper you recommended in our related work and cited it.
Secondly, we have restructured our paper to follow the standard format of Introduction-Related Works-Methodology-Experiments and Results analysis-Discussion-Conclusion. We have also improved the readability and clarity of each section to ensure that our paper is more coherent and easier to understand.
Thirdly, we have added a discussion section to provide a more comprehensive comparison of our method with the existing methods. We have highlighted the strengths and weaknesses of our approach, as well as those of other state-of-the-art methods, to help readers better understand the performance of different methods.
Fourthly, we have rewritten the conclusion section to emphasize the main contributions and implications of our work in terms of accuracy, efficiency, robustness, and generalization. We have also addressed how our research can be applied in other contexts and how it can be improved by further work. We have made sure that the conclusions are not just a repetition of the results but are insightful and informative.
Finally, we have integrated and highlighted the strengths and weaknesses of our proposed method, referring to a comparison with the state of the art in the discussion section. We believe that this correction substantially increases the interest of the remote sensing community in our work.
We hope that these revisions have addressed your concerns and improved the quality of our paper. We thank you again for your time and effort in reviewing our paper. We look forward to hearing from you soon.
Sincerely,
Hawking Lai and Eddie Law
Reviewer 2 Report
In this paper, authors design a novel network model of object classification. This model replace the MLP layers in the existing PointMLP model to form the GhostMLP model according to the advantages of the GhostNet design. The experiments results show that the model can achieve better classification performance with lower computation. Especially, Compared with PointMLP, the proposed method maintains sustainable performance with fewer parameters and lower FLOPs computations. The classification results show that the proposed model is superior to the PointMLP on the ScanObjectNN dataset, achieving 88.7% overall accuracy and 87.6% mean accuracy. Additionlly, authors also introduces another lightweight version GhostMLP-S. These two models completes faster training and inference with GPU and is the best-performing method that does not require any extra training data in the ScanObjectNN benchmark. The innovation of this paper is enough and interesting. However, a few things are to be considered sincerely before its publication.
1. The topic of this paper is about Object Classification or segmentation, in the introduction and related works, authors only introduces the deep learning method in the field of Classification or segmentation, lacks clusteing algorithm, especially the latest clustering algorithm in the application of classification, such as exponential function density clustering, density peak clustering .
Chen X, Wu H, Lichti D, et al. Extraction of indoor objects based on the exponential function density clustering model[J]. Information Sciences, 2022, 607: 1111-1135.
A. Rodriguez and A. Laio. Clustering by fast search and find of density peaks, Science, 344(6191) (2014) 1492-1496.
2. Line 221: In this section, the performance benchmarks of the GhostMLP models for 3D shape classifications on three datasets. The content include classification and Part segmentation, therefore, the description should be rigorous.
3. in the section of “4.3. Part segmentation on ShapeNet”, authors introduce the experiment of part segmentation. The topic is “Object Classifications”, therefore ,we suggest the topic of the paper should be changed about Classification and segmentation.
4. The 4.3 section “Part segmentation on ShapeNet” should include some picture about the part segmentation results.
5. Table 1 shows that the time efficiency, the time efficiency of Classification is the same with Part segmentation?
Author Response
Dear Reviewer #2,
Thank you for providing us with your valuable comments and suggestions on our paper. We sincerely appreciate your constructive feedback and interest in our work. We have taken into consideration all your recommendations and have carefully revised our paper accordingly.
Deep learning approaches have carried out the point cloud classifications. Using the exponential kernel function for data clustering is another feasible approach as recommended by you. Therefore, we added the reference and stated in our paragraph (page 1, lines 26-28) that there are other possible methods also in point cloud data classifications.
Regarding the textual mistake on line 221, we have corrected this error and made sure to carefully proofread our manuscript for any other potential errors.
Regarding your comments on section 4.3 on part segmentation, we think that part segmentation in the point cloud can be considered as a kind of classification. In part segmentation, each point in the point cloud is classified into different parts or segments based on its properties or features. Therefore, we have decided to keep the topic of the paper as "Classification," as it encompasses both classification and segmentation tasks.
We have also included pictures of the part segmentation results and added more descriptions about speed/efficiency (Table 1 for classification, line 346 for part segmentation, line 382 for large-scale point cloud classification) for part segmentation and other experiments. These changes should help point to cloud analysis and demonstrate the effectiveness of our proposed GhostMLP model in those tasks.
We hope that our revisions have addressed all of your concerns and improved the quality of our manuscript. Once again, we appreciate your feedback and are open to further comments or suggestions.
Thank you for your time and consideration.
Best regards,
Hawking Lai and Eddie Law
Reviewer 3 Report
Authors should supplement the units of some data in the table. such as speed.
Author Response
Dear Reviewer #3,
Thank you for your valuable feedback on our paper. We appreciate the time and effort you have put into evaluating our work.
We have carefully reviewed your comments and agree that the units of some data in the tables should be supplemented. Specifically, we will add units to the speed data to provide a clearer understanding of the results. You could do a quick check using the track change version.pdf.
We appreciate your attention to detail and thank you for bringing this to our attention. If you have any further suggestions or feedback, please do not hesitate to let us know.
Thank you again for your review.
Best regards,
Hawking Lai and Eddie Law.
Round 2
Reviewer 2 Report
All comments have done.